# Log BB Prediction Models Using TLC and HPLC Retention Values as Protein Affinity Data

**DOI:** 10.3390/pharmaceutics16121534

**Published:** 2024-11-30

**Authors:** Karolina Wanat, Klaudia Michalak, Elżbieta Brzezińska

**Affiliations:** Department of Analytical Chemistry, Faculty of Pharmacy, Medical University of Lodz, 90-419 Lodz, Poland; klaudia.michalak@stud.umed.lodz.pl (K.M.); elzbieta.brzezinska@umed.lodz.pl (E.B.)

**Keywords:** log BB, CNS penetration, regression models, chromatographic data, affinity chromatography, serum albumin

## Abstract

Background: The penetration of drugs through the blood–brain barrier is one of the key pharmacokinetic aspects of centrally acting active substances and other drugs in terms of the occurrence of side effects on the central nervous system. In our research, several regression models were constructed in order to observe the connections between the active pharmaceutical ingredients’ properties and their bioavailability in the CNS, presented in the form of the log BB parameter, which refers to the drug concentration on both sides of the blood–brain barrier. Methods: Predictive models were created using the physicochemical properties of drugs, and multiple linear regression and a data mining method, i.e., MARSplines, were used to build them. Retention values from protein-affinity chromatography (TLC and HPLC) were introduced into the analyses. In both experiments, the stationary phases were modified with serum albumin, which enriched the obtained chromatographic data, and were then introduced into the models with good results. Results: The conducted analyses confirm that the variables that influence the log BB include high degree of lipophilicity, ionisation capacity and low capability of forming hydrogen bonds. However, the addition of chromatographic data improved the obtained regression results and increased the robustness of the models against an increased number of cases. The linear regression model with chromatographic parameters explains 85% of the log bb variability, whereas the MARSplines model explains 91%. **Conclusions:** Based on the obtained results, it can be concluded that the use of chromatographic data can increase the robustness of predictive regression models related to penetration through biological barriers.

## 1. Introduction

### 1.1. BBB Permeability Indices in Prediction Models

For many years, various scientists have focused on predicting the penetration of xenobiotics through the blood–brain barrier, the BBB [1,2,3], which is a key pharmacokinetic property of centrally acting drugs or potential drug structures. Despite the development of available technology in the process of designing new medicinal substances and a deeper understanding of the phenomena underlying many neurological diseases, introducing to the market a new drug with high effectiveness in the central nervous system (CNS) remains a major challenge. This is due to the presence of physiological barriers separating cerebral circulation from peripheral circulation, thus protecting the brain from access to potentially toxic chemicals. The difficulty in overcoming these barriers for drugs results from the presence of tight junctions between cerebral capillary endothelial cells, which prevent intercellular transport from the blood to the brain. Another obstacle is the system of efflux transporters, such as P-glycoprotein (P-gp) or organic anion transporting polypeptides (OATPs), which remove metabolic residues from the interstitial fluid of nervous tissue [4,5,6].

In order to quickly assess the bioavailability in the CNS, a number of relatively easy computational methods have been developed, all in order to maximize the effect and reduce the costs and duration of the study. In silico models were successfully used to estimate the degree of BBB penetration using the influence of various physicochemical properties of the tested chemical compounds on their distribution in the body. The most frequently analyzed molecular descriptors (MDs) concern the size of molecules, their hydrophilic properties and the degree of ionisation at a physiological pH [7].

With this type of research method, it is necessary to reduce the studied phenomenon to some kind of measurable value. Various parameters have been introduced, which serve as variables reflecting the phenomenon of substance penetration through the BBB. It started with log BB, which is the ratio between the drug concentration on both sides of the BBB and log PS, which is the permeability of the surface-area product [8]. Over time, more complex parameters were introduced, such as the total brain-to-plasma drug partition coefficient (K_p_), the total brain-to-unbound plasma partition coefficient (K_p,u_) and its most developed form, K_p,uu,brain,_ which stands for unbound brain-to-plasma drug partition coefficient, which describes only the concentration of the unbound drug in the brain relative to the blood at a steady state and is determined by the clearance of influx and efflux [9]. K_p,uu,brain_ is usually obtained using animal models, but attempts have, also been made to build a predictive model [9].

Table 1 represents BBB penetration models from the last 5 years. In recent years, more emphasis has been placed on data mining techniques. They consist of searching for dependencies between variables in datasets. Machine learning is often used to build models, which ensures a good model fit, but requires good quality data and as many cases as possible used in the model. More traditional attempts, like multiple linear regression (MLR), are limited due to the one-dimension algorithm and they can serve as preliminary research to observe the correlation between variables and their impact on the predicted parameter.

### 1.2. Chromatography in QSAR Modelling

To enhance the possibilities of in silico prediction, scientists introduce various parameters to enrich the models, including parameters that are usually unrelated to the molecular characteristics of xenobiotics. Separation methods such as liquid chromatography (HPLC and TLC) are used as a data source for biological barrier penetration models. In a review in 2022, Ciura and Dziomba meticulously collected and described examples of the use of liquid chromatography in BBB penetration studies [19]. The possibility of using different partition mechanisms between two chromatographic phases allows us to create experimental conditions as close as possible to the physiological environment (of course, this similarity is limited due to the possibilities of such experiments). Modified stationary phases, including those imitating biological membranes, immobilized artificial membranes (IAM), or mobile phases with buffers or surfactants, as in micellar liquid chromatography (MLC), are just one of many examples.

The binding of drugs to serum proteins is a key pharmacokinetic aspect of crossing biological barriers, including the BBB. Due to its high plasma concentration, nonspecific binding and high affinity, albumin is the main blood transport protein. One of the most frequently used techniques in the study of biologically active compounds is high-performance affinity chromatography (HPAC). It uses highly specific interactions between ligands adsorbed on the surface of the stationary phase and specific components of the mixture being separated. In the presented studies, chromatography was used in thin-layer and column formats, in which the affinity of drugs for albumin was determined. For this purpose, the stationary phases were modified with the following serum albumins: bovine serum albumin (BSA) was used in TLC; human serum albumin (HSA) was used in HPLC experiments. Retardation factors (R_f_) along with the retention coefficient (log k) were analyzed and the data obtained on the basic pharmacokinetic parameters of the tested compound were included in the construction of the model.

### 1.3. Log BB

log BB is defined as the decimal logarithm of the ratio of the drug concentration in the brain to the drug concentration in the blood under the state of dynamic equilibrium of the drug distribution process between the blood and the brain. This value is based on a measurement of the substance concentration in the brain and blood after intravenous administration. However, this approach is associated with a high risk of error because it largely depends on the time of measurement. The determination of concentration–time curves provides much more valuable data on the log BB value, but is also much more expensive and time-consuming. The disadvantage of log BB is that it does not take into account pharmacokinetic aspects such as protein binding. Therefore, drugs that are highly bound to plasma proteins may have a low log BB value. For some compounds, this parameter may reach a value less than zero. Nevertheless, these drugs cross the blood–brain barrier in an unbound state and act centrally (e.g., ibuprofen or midazolam) [8,20]. The usefulness and advantage of the log BB parameter over other BBB indicators lie in the ease of its determination compared to the determination of kinetic parameters of blood–brain barrier penetration and a wide range of possibilities of finding log BB values in the literature (in comparison with K_p,uu,brain_) or using online calculators.

log BB limits vary between prediction models. The optimal classification threshold is usually set between 0 and −1 (see Table 1). One source states that penetrating compounds (BBB+) are characterized by a log BB value above 0. For compounds that do not penetrate the blood–brain barrier (BBB−), some time ago, a limit value of log BB of < −0.3 was established, but it is the value of log BB = −0.52 that seems to be the logical division into BBB+ and BBB−. This is equivalent to a 30% ratio of the concentration of the compound in the brain to the concentration in plasma. Our previous research established another limit of log BB ≥ −0.9 as the one useful in calculations [21,22,23].

The experiment provided in this work concerns the relationship between the ability of drugs to penetrate the central nervous system and their affinity to form connections with proteins; the correlation matrix presented below (Table 2) includes the log BB (indexed as LOGBB) parameter together with its limits described earlier, i.e., −0.9 and −0.52, CNS penetration ability (CNS+/−) and protein binding extent (PB). These parameters were adopted based on the bibliography [7,21,22,23]. LOGBB is quantitative here, corresponding to log BB = 0.547−0.016 PSA [24].

The correlation matrix indicates a favorable reflection of BBB penetration by LOGBB and LOGBB > −0.9 indices. The apparent relationship between PB and the ability to cross the blood–brain barrier is inversely proportional. This observation result was expected and this relationship is quite strongly correlated (R = −0.41 and −0.45). On this basis, the further course of the experiment was envisaged. The behavior of the drug in the human body and in the experimental environment depends on their physicochemical properties. These relationships are directly or inversely proportional; however, a limited group of cases is always considered. Thus, the range of variability of individual physicochemical features has a minimum, maximum and average. This affects the obtained mathematical model of the dependence of drug behavior on these features. A model created on the basis of one group of cases may have a different form than that created on the basis of another group. The variance in the descriptor value in a group of cases can even change the correlation from positive to negative and vice versa. It is always beneficial to determine the peak of a given feature.

## 2. Materials and Methods

### 2.1. Thin-Layer Chromatography

The chromatographic data for the dataset of 181 active pharmaceutical ingredients, including APIs (initially assembled working group of 37 drugs, which was then expanded to a dataset of 181 APIs), were collected in the following manner: 10 µL of 1 mg/mL API solutions in gradient-grade methanol was applied on normal-phase and reversed-phase TLC plates and air-dried. The plates were then developed in the mobile phase composed of acetonitrile, methanol and acetic buffer at pH 7.4, 60:20:20 (*v*/*v*/*v*), respectively. After drying, the plates were subjected to densitometric scanning with the analytical wavelength differing between 220 and 300 nm and the retardation factors for each API were collected. Then, the experiment was repeated but with stationary phases modified with bovine serum albumin (BSA); the plates were earlier covered with 2 mg/mL aqueous solution of BSA and air-dried. The collected R_f_ were indexed as “NP” as the R_f_ collected from the normal-phase BSA-modified plate and “RP” for the R_f_ from the reversed-phase BSA-modified plate.

### 2.2. High-Performance Liquid Chromatography

The HPLC experiment was performed on two columns with the following modified stationary phases: HSA (immobilized human serum albumin) and IAM (immobilized artificial membrane). The mobile phases were composed of PBS buffer at pH 7.4, acetonitrile and methanol (85:10:5, *v*/*v*/*v*) for the HSA column; for the IAM column, it was composed of PBS buffer at pH 7.4 and acetonitrile (80:20, *v*/*v*). Detection was performed using a UV–VIS spectrometer, with the analytical wavelength of 210 nm for each API.

Chromatographic data (retention coefficient k and derivative log k) were obtained from both HPLC experiments. The final results were the average of two repetitions of the experiment, and they were noted as logk_HSA_ and logk_IAM_, respectively.

### 2.3. Data Collecting and Computational Experiments

Physicochemical properties of all APIs were calculated in Hyperchem software (HyperChem for Windows Release 7.02, HyperCube Inc., Gainesville, FL, USA, 2002). Physiological properties such as protein binding, acid/base nature or CNS availability (Table 3) were collected using the following online databases: DrugBank [25] and Chembl [26]. The data for 181 APIs are available in Appendix A. All statistical analyses were performed using STATISTICA 13.3 software (TIBCO Software Inc., Palo Alto, CA, USA, 2021).

## 3. Results

### 3.1. Multiple Regression Models Using Physicochemical Parameters

The analysis started with the small dataset of 37 API to determine the trends that occur between variables. All CNS bioavailability indices, including LOGBB, CNS+/−, LOGBB > −0.52 and LOGBB > −0.90, are correlated with physicochemical descriptors and the degree of protein binding. This is presented on the correlation matrix (Table A1, Appendix B) and based on this, a set of regression summaries has been calculated (Equations (1)–(4)).
LOGBB = 0.85(± 0.18) − 0.14(± 0.01)HA + HD − 0.77(± 0.23)PBR^2^ = 0.84; F(2.32) = 83.399; *p* < 0.00000; s = 0.25278O^2^_LOO_ = 0.81; PRESS = 3.528; S_PRESS_ = 0.3174; SDEP = 0.1818; Q^2^_LMO_ = 0.80(1)
LOGBB > −0.52 = −0.35(± 0.44) − 0.26(± 0.03)HA + 0.15(± 0.05)PB_code_ − 0.09(± 0.04)eH–eL + 0.28(± 0.07)MW − 0.10(± 0.03)logDR^2^ = 0.80; F(5.29) = 23.671; *p* < 0.00000; s = 0.23876(2)
LOGBB > −0.9 = 1.70(± 0.28) − 0.17(± 0.04)HA − 0.81(± 0.28)PB − 0.09(± 0.04)log DR^2^ = 0.56; F(4.30) = 9.4142; *p* < 0.00005; s = 0.31435(3)
CNS+/− = 1.05(± 0.22) − 0.11(± 0.02)HA − 0.08(± 0.03)DM + 0.12(± 0.06)PB_code_R^2^ = 0.60; F(3.31) = 15.494; *p* < 0.00000; s = 0.31201(4)

The regression summary of the dependent variable LOGBB yielded the best result (Figure 1a). The resulting equation explains 84% of the variance of the LOGBB. This variable is a set of quantitative values, where the other indices, CNS+/−, LOGBB > −0.52 and LOGBB > −0.90, are coded values, i.e., −1 for cases that cross the blood–brain barrier and 0 for cases that do not. Thus, the scatterplot for the other models cannot be determined. They successively explain 60%, 80% and 56% of the variance of the volatility of these indicators.

In all the regression summaries of the tested indices, i.e., LOGBB, CNS+/−, LOGBB > −0.52 and LOGBB > −0.90, there are descriptors connected to the number of possible hydrogen bonds formed by the analyzed drugs (HA—acceptors; HA + HD—sum of acceptors and donors) and their connections with proteins (PB and PB_code_). In the case of LOGBB > −0.52 and LOGBB > −0.90, the following descriptors also appear: logD and MW, including hydro-lipophilic balance and molecular weight. The indicator LOGBB > −0.52, which is the most widely represented, also changes with the electronic parameter eH−eL, describing the possibility of ionisation of the drug.

The MLR was then repeated on the larger set of APIs (n = 181) with the independent variable LOGBB, which showed the best results with a small set of APIs. This time, the R^2^ was lower at 0.68 and the regression equation included HA again along with logP and logU/D (Equation (5)). The scatter plot is presented in Figure 1b.
LOGBB = 0.52(± 0.09) − 0.22(± 0.013)HA + (± 0.015)0.05logP + (± 0.019)0.04 logU/DR^2^ = 0.68; F(3148) = 109.20; *p* < 0.00000; s = 0.368O^2^_LOO_ = 0.62; PRESS = 2.324; S_PRESS_ = 0.2154; SDEP = 0.1287; Q^2^_LMO_ = 0.61(5)

The rest of the indices LOGBB > −0.9; LOGBB > −0.52 and CNS+/− resulted in coefficients of determination of 0.53; 0.52 and 0.38, respectively.

### 3.2. Chromatographic Data Analysis

The chromatographic data analysis showed that analytical models of drug binding to proteins best represent this phenomenon in the form of simple chromatographic data, including R_f_ from TLC and log k from HPLC (Table 4). Here, the research began immediately on an enlarged group of compounds, where n = 181. The highest correlations with the percentage of bound fraction showed that HPLC_HSA_: logk_HSA_ (0.61) and logk_IAM_ (0.36). In the case of TLC, the relevant correlation was observed with normal-phase R_f_ NP (0.25).

The best correlations 0.24–0.27 were again obtained for the LOGBB index with NP, RP and logk_HSA_, however, the logk_IAM_ variable stood out in particular with a correlation of 0.42. The presented correlations are also confirmed by the regression results; the best equation was achieved for the logk_IAM_ variable with a coefficient of determination of 0.88; similarly, high results were obtained for the NP and RP variables, with both equations explaining around 85% of the LOGBB variability (Equations (6)–(8)) (Figure 2).
LOGBB = 0.34(± 0.18) − 0.17(± 0.006)HA + HD − 0.24(± 0.07)NP − 0.039(± 0.02)eH−eL + 0.02(± 0.01)logP + 0.015(± 0.012)logU/DR^2^ = 0.86; F(5, 161) = 109.203; *p* < 0.00000; s = 0.243(6)
LOGBB = 0.53(± 0.19) − 0.17(± 0.006)HA + HD − 0.31(± 0.1)RP − 0.025(± 0.02)eH−eL − 0.01(± 0.009)DMR^2^ = 0.85; F(4, 162) = 241.06; *p* < 0.00000; s = 0.252(7)
LOGBB = 0.27(± 0.22) − 0.13(± 0.007)HA + HD − 0.01 (± 0.009)DM + 0.02(±0.01)logU/D − 0.04(±0.02)eH−eL + 0.0008(± 0.000)MW + 0.032logk_IAM_R^2^ = 0.88; F(6, 120) = 162.02; *p* < 0.00000; s = 0.132(8)

These results were then compared with the API working group n = 37, where regression equations were obtained only using NP and RP variables (this time, logk_IAM_ was not included in the model) and it was concluded that the use of TLC chromatographic data increased the robustness of MLR models against the occurrence of more cases (Table 5) compared to models that do not contain these data. Regression equations for n = 37 are available in Appendix B (Equations (A1) and (A2)).

### 3.3. Data Mining Models with Chromatographic Data

First, a set of 37 APIs was analyzed using the MARSplines method, which is a non-parametric regression algorithm. The variables used in model building were the same as those that occur in multiple linear regression models. A basic model with physicochemical properties only was created using HA; logP with quantitative and PB_code_ as the qualitative predictor. The coefficient of determination was R^2^ = 0.88 and the standard deviation of prediction was s = 0.2426. Introduction of TLC chromatographic data (variable NP) improved the results (Table 6). The best model with R^2^ = 0.91 is presented in Figure 3.

The presence of the variable NP significantly increased the correctness of the model fit (0.88 to 0.91) and reduced the prediction error (0.2426 to 0.2076). Retention values from HPLC (logk_HSA_ or logk_IAM_) were below the significance level; therefore, no models were built with these data.

To test the robustness of the model with an increasing number of cases, the second model was built using data from 181 APIs (Figure 4). The coefficient of determination achieved lower values, which was the expected result, but it still represents decent predictive power (R^2^ = 0.76; s = 0.3180).

## 4. Discussion

The statistical analyses carried out in the group of 37 cases highlighted the physicochemical properties such as low molecular weight (MW), high lipophilicity (logD) and low hydrogen bonding capacity (HA, HD; HA + HD). These physicochemical parameters, related to Lipinski’s “rule of five”, confirm their association with the BBB bioavailability. HA and HD had a particularly large impact on the LOGBB parameter, i.e., the ratio of the drug concentration in the CNS to its concentration in the blood; this was the variable that determined most of the success in fitting the model to the trend line. Therefore, the main goal of the analyses was to assess the presence of other independent variables in the regression equations and to check the model’s resistance to an increase in the number of cases. This task was undertaken because the common result is a high R^2^ for small datasets, which drops dramatically for models of larger sets of APIs; it also helps to confirm the reliability of the model when initially, with small datasets, it may be possible to select cases where a good fit was achieved. Such a situation also occurred with the model for 37 compounds, which explained as much as 84% of the LOGBB variability, while the model for an enlarged group of compounds (n = 181) explained only 68%.

The second important aspect that was featured in the working group of 37 APIs was the PB parameter, i.e., binding to plasma proteins. It was present in each model for LOGBB, LOGBB > −0.52, LOGBB > −0.90 and CNS+/−. This resulted in an attempt to implement chromatographic data from plasma albumin-modified stationary phases from TLC and HPLC experiments into multiple linear regression models. These attempts were successful, and the parameters NP and RP from TLC chromatography were entered into the regression equations instead of PB with good results. Both models explained around 85% of the LOGBB variability in the model for a large number of cases (n = 181). Interestingly, when using NP or RP parameters instead of PB, resistance against an increase in the number of APIs in the model was achieved. For n = 37, the R^2^ values were only slightly higher than for an enlarged dataset and were 0.90 and 0.88, respectively (see Table 5).

Analysis of the chromatographic data proved the connection of these parameters to the binding to plasma proteins. The highest correlations with the PB were logk_HSA_ and NP; however, the retention factors obtained from high-performance liquid chromatography with the immobilized human albumin (logk_HSA_) column contributed less to the regression models.

It was also decided to examine chromatographic data from a column with an immobilized artificial membrane, the logk_IAM_ parameter, i.e., retention coefficient data from an artificial membrane column, which helped us to assess the permeation of substances through biological membranes. In the multiple linear regression model for a large group of compounds where n = 181, the highest result so far was obtained when R^2^ = 0.88, which could also be predicted by looking at the correlation matrix because logk_IAM_ demonstrated 42% correlation with LOGBB. Data mining models confirmed the strong contribution of TLC data with the MARSplines method. The HPLC data were below statistical significance and did not enter the models. The robustness of the regression model against the number of APIs was lower than in the case of MLR and the coefficient of determination decreased from 0.91 to 0.76

## 5. Conclusions

It can be concluded that although the blood–brain barrier is an extremely complex structure, with many mechanisms protecting against the penetration of xenobiotics into the CNS structures, transport is still largely based on the principle of penetration through biological membranes. Plasma protein binding, which limits drug distribution, also seems to be of great importance in regression models; therefore, chromatographic data related to protein binding can be useful in such predictions. This applies primarily to retention factors from thin-layer chromatography in both normal- and reverse-phase (NP and RP) systems, which were present in each regression model (both MLR and MARSplines), and their particularly positive impact on the robustness of the multiple linear regression model could be seen when increasing the number of cases from 37 to 181.

The obtained results justify further development of the use of chromatographic data in predictive models. The undoubted advantage is the high universality of the obtained models, which are applicable to drugs from various chemical groups and with different acid–base natures, as well as the simplicity and low cost of conducting TLC and HPLC chromatographic experiments. The use of chromatographic data can be extended to other biological barriers because the separation of compounds between the two phases (solid and mobile) can be used to map the basic mechanisms of drug penetration through such barriers in vitro.

## Figures and Tables

**Figure 1 pharmaceutics-16-01534-f001:**
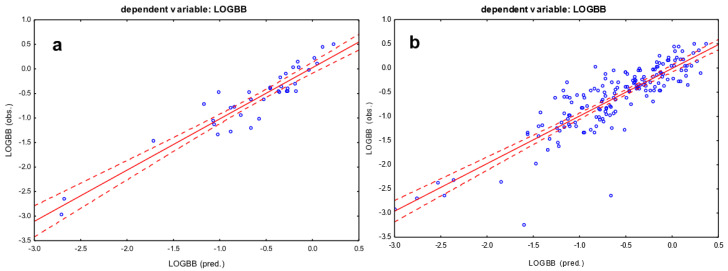
Scatter plot of the multiple linear regression model from Equation (1) (**a**) and Equation (5) (**b**). Dependent variable: LOGBB; coefficient of determination: R^2^ = 0.84 for (**a**) and R^2^ = 0.68 for (**b**). The confidence interval 0.95 is marked on the chart (red dotted lines).

**Figure 2 pharmaceutics-16-01534-f002:**
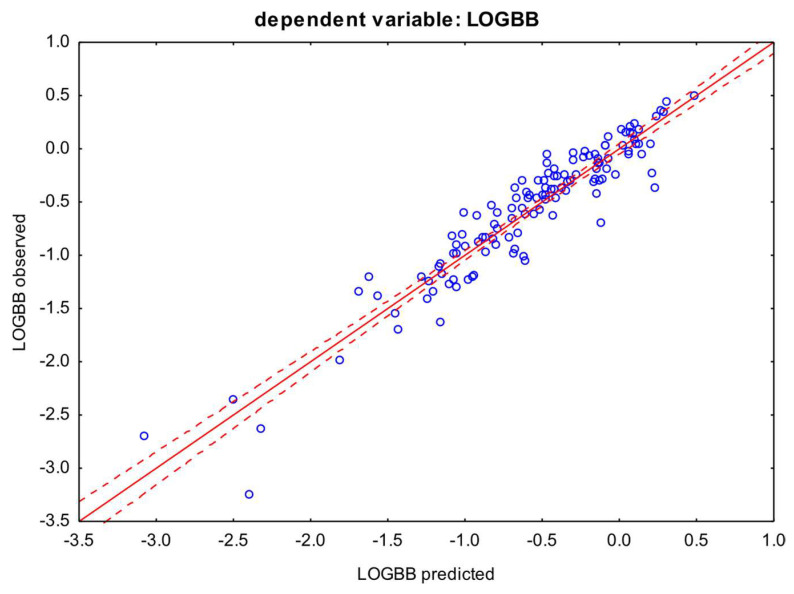
Scatter plot of the multiple linear regression model from Equation (8) obtained with logk_IAM_. Dependent variable: LOGBB. Coefficient of determination: R^2^ = 0.88. The confidence interval 0.95 is marked on the chart (red dotted lines).

**Figure 3 pharmaceutics-16-01534-f003:**
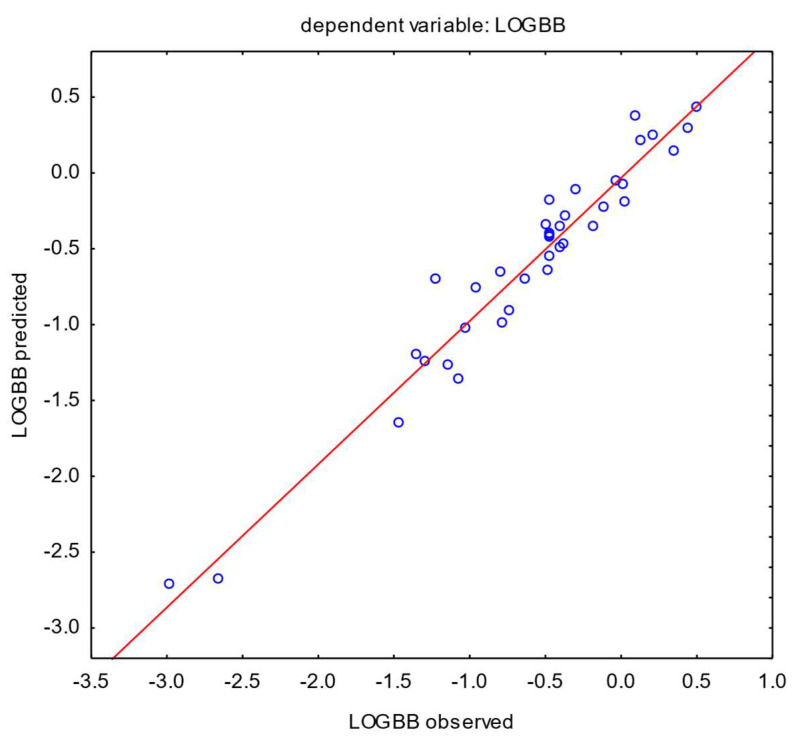
MARSplines model for 37 active pharmaceutical ingredients. Five basic functions were used in the model algorithm. Coefficient of determination: R^2^ = 0.91.

**Figure 4 pharmaceutics-16-01534-f004:**
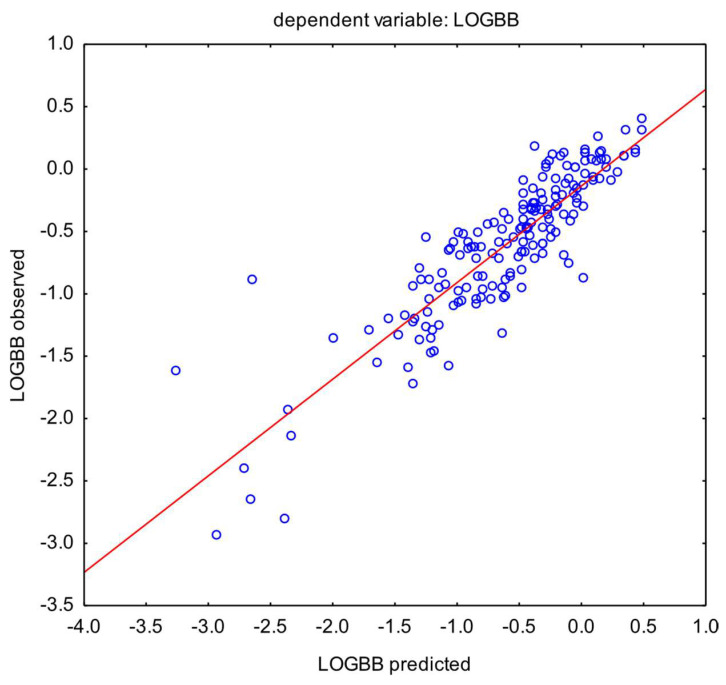
MARSplines model for 181 active pharmaceutical ingredients. Seven basic functions were used in the model algorithm. Coefficient of determination: R^2^ = 0.76. Regression line marked in red.

**Table 1 pharmaceutics-16-01534-t001:** Blood–brain barrier penetration models from the last 5 years.

Statistical Method	Parameter	No. of Compounds in the Model	Prediction Power of the Model	Reference
BBB Score model (Python script)	CNS and non-CNS	990 in total (270 CNS and 720 non-CNS)	BBB Score area under the curve AUC = 0.86	Gupta 2019 [10]
Light Gradient Boosting Machine (LightGBM)	BBB+ and BBB−	7162 in total (5453 BBB+ and 1709 BBB−)	LightGBM model accuracy = 89%	Shaker 2020 [11]
Artificial neural network (ANN)	log BB	529	Q^2^ = 0.815 RMSE_cv_ = 0.318	Radchenko 2020 [12]
Random forest (RF) and a MACCS molecular fingerprint	BBB+ and BBB−	1593 in total (1283 BBB+ and 310 BBB−)	RF model accuracy = 91%	Liu 2021 [13]
Deep learning (DNN, CNN-1D) Machine learning (SVM, kNN, RF, and NB)	BBB+ and BBB−	3605	DNN model accuracy = 98.07% 1-D CNN model accuracy = 97.44% CNN by transfer learning model accuracy = 97.61%	Kumar 2022 [14]
QSAR softwares: Leadscope Enterprise (LS) and CASE Ultra (CU)	log BB	921	LS_cv_ model accuracy = 80% CU_cv_ model accuracy = 86%	Faramarzi 2022 [15]
XGboost, RF, Extra-tree classifiers and DNN	BBB+ and BBB−	8153	DNN model accuracy = 97.8%	Mazumdar 2023 [16]
logBB_Pred tool—LightGBM	logBB (quantitative model) BBB+ and BBB− (qualitative model)	1276	log BB prediction model accuracy = 61% BBB+/− prediction model accuracy = 85%	Shaker 2023 [17]
RF, SVM, GBM and NN	BBB permeability (quantitative model)	307	R^2^ of 2 subsequent models: RF = 87%, 74% NN = 87%, 59% GBM = 85%, 80% SVM = 78%, 52%	Yang 2024 [18]
ANN, MLR, SVM, PLS	K_p,uu,brain_	256 plus 32 for external validation	ANN model accuracy = 86.7%	Ma 2024 [9]

**Table 2 pharmaceutics-16-01534-t002:** Drug bioavailability indices and their relationship with drug binding to proteins in the correlation matrix.

n = 37	CNS+/−	LOGBB > −0.9	LOGBB	LOGBB > −0.52	PB
CNS+/−	1.00				
LOGBB > −0.9	0.36	1.00			
LOGBB	0.57	0.77	1.00		
LOGBB > −0.52	0.53	0.78	0.73	1.00	
PB	−0.18	−0.41	−0.45	−0.26	1.00

**Table 3 pharmaceutics-16-01534-t003:** Independent variables (physicochemical, pharmacokinetic and chromatographic) used in regression models.

Variable	Description
Acid/base	Acidic (a), basic (b) or neutral (n) nature of an API
LOGBB	Penetration through the blood–brain barrier = 0.547−0.016 PSA
CNS+/–	Bioavailability in the CNS
eL–eH	Ionisation capacity
HA	Hydrogen bond acceptors
HD	Hydrogen bond donors
log D	Distribution coefficient
log k_HSA_	Logarithm of retention factor from HPLC_HSA_
log k_IAM_	Logarithm of retention factor from HPLC_IAM_
log P	Partition coefficient
log U/D	Extent of ionisation
MW	Molecular weight
NP	The R_f_ from NP BSA–modified plate
PB	Protein binding
PB_code_	PB converted to categorical values (code: 1–5)
PhCharge	Charge of a compound in physiological environment
PSA	Polar surface area of a molecule
RP	The R_f_ from RP–2 BSA–modified plate
Sa	Surface area of a molecule
V	Molecular volume

**Table 4 pharmaceutics-16-01534-t004:** Correlation matrix of BBB penetration indexes, including CNS+/−; LOGBB; LOGBB > −0.9; LOGBB > −0.52, with the following chromatographic data: variables NP and RP from TLC; variables logk_HSA_ and logk_IAM_ from HPLC.

n = 181	CNS+/−	LOGBB	LOGBB > −0.9	LOGBB > −0.52	PB	NP	RP	logk_HSA_	logk_IAM_
CNS+/−	1.00								
LOGBB	0.61	1.00							
LOGBB > −0.9	0.65	0.81	1.00						
LOGBB > −0.52	0.57	0.78	0.75	1.00					
PB	0.23	0.29	0.17	0.14	1.00				
NP	−0.065	−0.24	−0.17	−0.22	0.26	1.00			
RP	0.017	−0.25	−0.16	−0.20	−0.14	0.52	1.00		
logk_HSA_	0.15	0.27	0.15	0.19	0.62	−0.037	−0.21	1.00	
logk_IAM_	0.29	0.42	0.39	0.29	0.37	−0.22	−0.35	0.30	1.00

**Table 5 pharmaceutics-16-01534-t005:** The results of MLR models on working group of APIs (n = 37) compared to enlarged dataset of 181 APIs.

No. of Cases	n = 181	n = 37
NP TLC	R^2^ = 0.86	R^2^ = 0.90
RP TLC	R^2^ = 0.85	R^2^ = 0.88
logk_IAM_	R^2^ = 0.88	−

**Table 6 pharmaceutics-16-01534-t006:** MARSplines prediction models with TLC retention values (n = 37 APIs).

MARSplines Model	NP TLC	RP TLC
Predictors in the model:	HA, logP, NP	HA, logP, RP
statistics	R^2^ = 0.91 s = 0.2076	R^2^ = 0.89 s = 0.2438
MARSplines model	Without TLC data	
Predictors in the model: statistics	HA, logP, PB_code_ R^2^ = 0.88 s = 0.2426	

## Data Availability

The original contributions presented in the study are included in the Appendix A, further inquiries can be directed to the corresponding author/s.

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
