# Peer review of "Log BB Prediction Models Using TLC and HPLC Retention Values as Protein Affinity Data"

_pharmaceutics, 2024, doi:10.3390/pharmaceutics16121534_

Round 1
Reviewer 1 Report
Comments and Suggestions for Authors
The manuscript describes an interesting prediction study for the passage of compounds in the brain through blood-brain barrier (BBB) using the parameter LOGBB corresponding to the decimal logarithm of the ratio of the drug concentration in the brain to the drug concentration in the blood, using protein affinity chromatography.
Although the presented work is of interest, the manuscript should be revised for clarity.
1) the title is not informative enough to correspond to the article content
2) please avoid abbreviations in the abstract, otherwise define them e.g. BBB, log U/D, etc Even if the article is for spécialistes, please take into account that non spécialistes could read it. The parameter log BB is defined lines97-98. When presenting an abbreviation for the first time use brackets to introduce it e.g. blood-brin barrier (BBB). Homogenise the way introducing the abbreviations. Too many abbreviations hinder the message.
3) Homogenise the terme log BB, it appears in the text under several forms Log BB, log BB, LOGBB…
4) the introduction could be improved by adding information on what is BBB
5) define the abbreviations in the captions of figures.
6) results section: what is the rationale of presenting both R and R2 for regression ?
7) the final paragraph with conclusions should be developed to give a better view for the perspectives of this work.
Author Response
Reviewer 1: The manuscript describes an interesting prediction study for the passage of compounds in the brain through blood-brain barrier (BBB) using the parameter LOGBB corresponding to the decimal logarithm of the ratio of the drug concentration in the brain to the drug concentration in the blood, using protein affinity chromatography.
Although the presented work is of interest, the manuscript should be revised for clarity.
Reply to the Reviewer 1: Thank you for all your suggestions and feedback. I refer to individual comments below
- the title is not informative enough to correspond to the article content
The title has been changed to “log BB prediction models using TLC and HPLC retention values as protein affinity data”
- please avoid abbreviations in the abstract, otherwise define them e.g. BBB, log U/D, etc Even if the article is for spécialistes, please take into account that non spécialistes could read it. The parameter log BB is defined lines97-98. When presenting an abbreviation for the first time use brackets to introduce it e.g. blood-brin barrier (BBB). Homogenise the way introducing the abbreviations. Too many abbreviations hinder the message.
Reply: The abstract has been rebuilt according to the editor's recommendation: Background/Methods/Results/Conclusions. As suggested, unnecessary shortcuts have been removed.
- Homogenise the terme log BB, it appears in the text under several forms Log BB, log BB, LOGBB…
Two entries, log BB and LOGBB, were introduced on purpose.
Log BB refers to the value of the ratio of drug concentrations in the blood and in the brain in theoretical terms and such a record appears in the introduction only. I suspect that the writing log BB and Log BB results only from the different location of this word in the sentence - if it was at the beginning of the sentence, it was written with a capital letter. To improve the clarity of the text, I changed all Log entries to log.
The LOGBB parameter was introduced to specify that the models presented refer to that specific mathematical value, as mentioned in the text:
„The experiment provided in this work concerns the relationship between the ability of drugs to penetrate the central nervous system and their affinity to form connections with proteins - the correlation matrix presented below (Table 2) includes log BB (indexed as LOGBB) parameter together with its limits described earlier: −0.9 and −0.52; CNS penetration ability (CNS+/−) and protein binding extent (PB). These parameters were adopted based on the bibliography [7,18–20]. LOGBB is quantitative here, corresponding to log BB = 0.547 – 0.016 PSA [21].”
- the introduction could be improved by adding information on what is BBB
A brief explanation of what the blood-brain barrier is can be found in the text:
“This is due to the presence of physiological barriers separating cerebral circulation from peripheral circulation, thus protecting the brain from access to potentially toxic chemicals. The difficulty in overcoming these barriers for drugs results from the presence of tight junctions between cerebral capillary endothelial cells, which prevent intercellular transport from the blood to the brain. Another obstacle is the system of efflux transporters, such as P-glycoprotein (P-gp) or organic anion transporting polypeptides (OATP), whose role is to actively remove toxic metabolic products from the interstitial fluid of nervous tissue [4–6]. “
I think that the introduction section is sufficiently extensive, and people interested in the topic presented have basic knowledge about the structure and operation of the blood-brain barrier. Inserting such construction descriptions or graphics may unnecessarily distract from the topic, especially since they are easily available in other sources.
- define the abbreviations in the captions of figures.
I replaced abbreviations with full names in all descriptions of figures
- results section: what is the rationale of presenting both R and R2for regression ?
As suggested, "R" values ​​have been removed from the equations and Table 5 to maintain text clarity
- the final paragraph with conclusions should be developed to give a better view for the perspectives of this work.
“Conclusions” section has been added to the manuscript:
“It can be concluded that although the blood-brain barrier is an extremely complex structure, with many mechanisms protecting against the penetration of xenobiotics into the CNS structures, the transport is still largely based on the principle of penetration through biological membranes. Plasma protein binding, which limits drug distribution, also seems to be of great importance in regression models, therefore chromatographic data related to protein binding can be useful in such predictions. This applies primarily to retention factors from thin layer chromatography in both normal and reverse phase (NP and RP) systems, that were present in each regression model (both MLR and MARSplines), and their particularly positive impact on the robustness of the multiple linear regression model could be seen when increasing the number of cases from 37 to 181.
The obtained results justify further development of the use of chromatographic data in predictive models. The undoubted advantage is the high universality of the obtained models, applicable to drugs from various chemical groups and with different acid-base nature, as well as the simplicity and economy of conducting TLC and HPLC chromatographic experiments. The use of chromatographic data can be extended to other biological barriers, because the separation of compounds between two phases - solid and mobile - can be used to map in vitro the basic mechanisms of drug penetration through such barriers.”
Reviewer 2 Report
Comments and Suggestions for Authors
The research aims to improve predictions of API bioavailability within the central nervous system, which is essential for drug design, especially in neuropharmacology. By analyzing the relationship between API properties and their CNS bioavailability, you offer valuable insights into enhancing drug efficacy and targeting. The approach is integration of protein-affinity chromatographic data, using both thin-layer (TLC) and high-performance liquid chromatography (HPLC) methods. The authors also performed statistical analysis. The constructive comment are as follows, which need to resolve before publication.
1. Abstract: Merge two paragraph, add concluding remark in abstract.
2. Referencing number correction: Line 30: [1–3][1,3,6], where is 4 and 5 reference. The reference should be like [1-4] and 6 will become 4.
3. Avoid bulk citation like above. Be specific and keep latest most relevant reference in place.
4. Line 60-61: Kp,uu,brain is obtained in vivo on animal models, but as the amount of data appears in 60 the literature, attempts are also made to build a predictive model. This need to cite.
5. Line 122, citation [84-86]. There is a serious issue of reference numbering. Please verify all numbers.
6. Line 149:,,, 181 active pharmaceutical ingredients…how this 149 drugs selected?
7. Why were TLC and HPLC specifically chosen, and were other protein-affinity chromatographic techniques considered? Is this based on literature or experience. Why https://doi.org/10.1016/j.biopha.2022.112828
8. 2.1, 2.2 etc methods need to cite.
9. How, Independent variables selected which presented in table 3?
10. It would be beneficial to clarify if the model's performance is consistent across various compound classes.
11. As serum albumin was used as the protein in these chromatographic phases, how do you anticipate this choice impacting model accuracy for drugs that interact with other proteins during BBB penetration?
12. Addressing some of the limitations and clarifying the model’s robustness across diverse datasets and chemical structures could further enhance the credibility and practical utility of the findings.
13. It would be useful if this prediction and discussion include the BCS.
Author Response
Reviewer 2:
The research aims to improve predictions of API bioavailability within the central nervous system, which is essential for drug design, especially in neuropharmacology. By analyzing the relationship between API properties and their CNS bioavailability, you offer valuable insights into enhancing drug efficacy and targeting. The approach is integration of protein-affinity chromatographic data, using both thin-layer (TLC) and high-performance liquid chromatography (HPLC) methods. The authors also performed statistical analysis. The constructive comment are as follows, which need to resolve before publication.
Reply to the Reviewer 2: Thank you for all your suggestions and feedback. I refer to individual comments below
- Abstract: Merge two paragraph, add concluding remark in abstract.
Reply: The abstract has been rebuilt according to the editor's recommendation: Background/Methods/Results/Conclusions. The conclusion is in one sentence as suggested in the review
- Referencing number correction: Line 30: [1–3][1,3,6], where is 4 and 5 reference. The reference should be like [1-4] and 6 will become 4.
Reply: This is my omission, [1,3,6] it was an older source marking that I forgot to remove, the same with the later one [84-86], thank you for pointing it out
- Avoid bulk citation like above. Be specific and keep latest most relevant reference in place.
Reply: The citation has been corrected and the number of references in the text has been reduced
- Line 60-61: Kp,uu,brain is obtained in vivo on animal models, but as the amount of data appears in 60 the literature, attempts are also made to build a predictive model. This need to cite.
Reply: The appropriate citation has been added (it refers to the last item in Table 1)
- Line 122, citation [84-86]. There is a serious issue of reference numbering. Please verify all numbers.
Reply: This citation has been corrected in [21-23], already explained in point 2
- Line 149:,,, 181 active pharmaceutical ingredients…how this 149 drugs selected?
Reply: The group of drugs to be tested has been collected for several years, these are commonly traded drugs, easily available and relatively easy to isolate, store, prepare solutions, etc. Their list and structures are available in the supplementary materials.
- Why were TLC and HPLC specifically chosen, and were other protein-affinity chromatographic techniques considered? Is this based on literature or experience. Why https://doi.org/10.1016/j.biopha.2022.112828
Reply: This results from the experience and current equipment capabilities in the Department. Additionally, both methods are simple and relatively quick to perform analyses.
- 2.1, 2.2 etc methods need to cite.
Reply: I don't understand the comment. Chromatographic experiments performed by us are described, so I wonder what to cite here?
- How, Independent variables selected which presented in table 3?
Reply: I understand that the question relates to how the independent variables in Table 3 were selected. As before: this is based on our experience of which variables are useful in studying the penetration of biological barriers that are mentioned in other articles. These are primarily the basic physicochemical or pharmacokinetic features (such as protein binding) that characterize chemical compounds.
- It would be beneficial to clarify if the model's performance is consistent across various compound classes.
Reply: The analyzes presented are basic for now, the topic will certainly be developed further. Thank you for your suggestion
- As serum albumin was used as the protein in these chromatographic phases, how do you anticipate this choice impacting model accuracy for drugs that interact with other proteins during BBB penetration?
Reply: The study was based on plasma albumin, because it is the main drug-binding protein in the bloodstream. Of course, a better reference to reality would be the entire fraction of plasma proteins, but this would make it difficult to perform chromatographic experiments. We are currently conducting research using acidic 1-alpha glycoprotein - AGP, and interestingly, the share of chromatographic data related to affinity for AGP is much lower
- Addressing some of the limitations and clarifying the model’s robustness across diverse datasets and chemical structures could further enhance the credibility and practical utility of the findings.
- It would be useful if this prediction and discussion include the BCS.
Reply to 12 i 13:
Again - the presented study shows preliminary results for now, further development of this topic is planned. The suggested improvements will certainly be taken into account in subsequent works
Round 2
Reviewer 1 Report
Comments and Suggestions for Authors
The authors taken into account most of the suggestions. The manuscript is now acceptable for publication